# Disrupted Calcium Signaling in Animal Models of Human Spinocerebellar Ataxia (SCA)

**DOI:** 10.3390/ijms21010216

**Published:** 2019-12-27

**Authors:** Francesca Prestori, Francesco Moccia, Egidio D’Angelo

**Affiliations:** 1Department of Brain and Behavioral Sciences, University of Pavia, 27100 Pavia, Italy; dangelo@unipv.it; 2Department of Biology and Biotechnology “Lazzaro Spallanzani”, University of Pavia, 27100 Pavia, Italy; francesco.moccia@unipv.it; 3IRCCS Mondino Foundation, 27100 Pavia, Italy

**Keywords:** spinocerebellar ataxias, Purkinje cells, Ca^2+^ signaling

## Abstract

Spinocerebellar ataxias (SCAs) constitute a heterogeneous group of more than 40 autosomal-dominant genetic and neurodegenerative diseases characterized by loss of balance and motor coordination due to dysfunction of the cerebellum and its efferent connections. Despite a well-described clinical and pathological phenotype, the molecular and cellular events that underlie neurodegeneration are still poorly undaerstood. Emerging research suggests that mutations in SCA genes cause disruptions in multiple cellular pathways but the characteristic SCA pathogenesis does not begin until calcium signaling pathways are disrupted in cerebellar Purkinje cells. Ca^2+^ signaling in Purkinje cells is important for normal cellular function as these neurons express a variety of Ca^2+^ channels, Ca^2+^-dependent kinases and phosphatases, and Ca^2+^-binding proteins to tightly maintain Ca^2+^ homeostasis and regulate physiological Ca^2+^-dependent processes. Abnormal Ca^2+^ levels can activate toxic cascades leading to characteristic death of Purkinje cells, cerebellar atrophy, and ataxia that occur in many SCAs. The output of the cerebellar cortex is conveyed to the deep cerebellar nuclei (DCN) by Purkinje cells via inhibitory signals; thus, Purkinje cell dysfunction or degeneration would partially or completely impair the cerebellar output in SCAs. In the absence of the inhibitory signal emanating from Purkinje cells, DCN will become more excitable, thereby affecting the motor areas receiving DCN input and resulting in uncoordinated movements. An outstanding advantage in studying the pathogenesis of SCAs is represented by the availability of a large number of animal models which mimic the phenotype observed in humans. By mainly focusing on mouse models displaying mutations or deletions in genes which encode for Ca^2+^ signaling-related proteins, in this review we will discuss the several pathogenic mechanisms related to deranged Ca^2+^ homeostasis that leads to significant Purkinje cell degeneration and dysfunction.

## 1. Introduction

The cerebellum, the second largest area exceeded only by the cerebral cortex, contains more neurons than the rest of the brain. For most of the 20th century, it has long been thought that the cerebellum was involved in motor learning and coordination [1,2]. More recently, it has been recognized to play a role in cognitive and emotional processes [3]. Thus, cerebellar dysfunctions can cause motor and/or non-motor deficits. Following lesions confined to cerebellum, Schmahmann and Sherman first described a novel clinical condition characterized by an impairment of (1) executive functions, (2) spatial cognition, and (3) language [4]. This constellation of symptoms, known in humans as cerebellar cognitive-affective syndrome (CCAS), was ascribed to the functional disruption of cerebellar connections to the association cortices and limbic system [4]. On the other hand, lesions to the cerebellum can cause abnormal and uncoordinated body movements, particularly of the gait (ataxia), tremor and loss of muscle tone [5,6]. There can be a myriad of causes for cerebellar damage, including cerebellar agenesis, dysplasia and hypoplasia, cerebellar stroke, tumors, inflammation, trauma, viral infections, toxin exposure, and neurodegenerative diseases, such as hereditary ataxias [7]. The hereditary ataxias are a group of genetically and clinically defined neurological disorders phenotypically characterized by gait ataxia, often accompanied by lack of coordination of hands, speech, and eye movements. Furthermore, atrophy of the cerebellum has frequently been detected [8,9]. Hereditary ataxia can be subdivided by mode of inheritance (i.e., autosomal dominant, autosomal recessive, X-linked, and mitochondrial), the mutated gene or chromosomal locus. Currently, more than 40 autosomal dominant types, named spinocerebellar ataxia (SCA) based upon the temporal order of discovery of the defective gene, are known, although the number is raising (Table 1) [10,11,12].

The most common forms of SCAs are caused by an abnormal expansion of the CAG (or CGT)-repeat sequence that leads to lengthening of polyglutamine (polyQ) tract in appropriate proteins [13,14,15,16]. In contrast, other forms of SCAs are characterized by conventional point mutations, including insertions and deletions [17,18,19]. Disruption of intracellular calcium (Ca^2+^) signaling in Purkinje cells is thought to be a key mechanism in the pathogenesis of SCAs [12,20,21,22]. Ca^2+^ signaling in Purkinje cells is important for normal cellular function as these neurons express a variety of Ca^2+^ channels, Ca^2+^-dependent kinases and phosphatases, and Ca^2+^-binding proteins to tightly maintain Ca^2+^ homeostasis and regulate physiological Ca^2+^-dependent processes. Abnormal Ca^2+^ levels can activate toxic cascades leading to cell death. The output of the cerebellar cortex is conveyed to the deep cerebellar nuclei (DCN) by Purkinje cells via inhibitory signals; thus, Purkinje cell dysfunction or degeneration would partially or completely impair the cerebellar output in SCAs. In the absence of the inhibitory signal emanating from Purkinje cells, DCN will become more excitable, thereby affecting the motor areas receiving DCN input and resulting in uncoordinated movements [22,23,24]. An outstanding advantage in studying the pathogenesis of SCAs is represented by the availability of a large number of animal models which mimic the phenotype observed in humans [6,25,26]. By mainly focusing on mouse models displaying mutations or deletions in genes which encode for Ca^2+^ signaling-related proteins, in this review we will discuss the hypothesis that the pathogenesis of diverse SCAs is related to deranged Ca^2+^ homeostasis that leads to significant Purkinje cell degeneration and dysfunction.

## 2. Mutations in the *CACNA1A* Gene

Different mutations in the *CACNA1A* gene, encoding for the pore-forming, voltage-sensing α1A-subunit of voltage-dependent Ca^2+^ Cav2.1 type channels (P/Q-type), are known to result in neurological disorders, such as episodic ataxia type 2 (EA2), familial hemiplegic migraine type 1 (FHM1) and SCA type 6 (SCA6; Table 1) [27,28,29]. Each disorder is associated with different mutations in the *CACNA1A* gene that have differential effects on Cav2.1 function and, therefore, either decrease or increase neuronal Ca^2+^ influx. SCA6 is associated with small CAG repeat expansions expressed as a polyglutamine (poliQ) sequence at protein level [28]. Voltage-dependent Ca^2+^ channels (VDCCs) mediate Ca^2+^ influx into neurons in response to membrane depolarization, thereby modulating cellular excitability and triggering a variety of Ca^2+^-dependent cellular processes, such as neurotransmitter release, synaptic plasticity, gene transcription, cell division and cell death [30,31]. P/Q-type VDCCs are highly expressed in the cerebellum, in particular in Purkinje cells where they account for more than 90% of Ca^2+^ currents [32,33,34,35]. P/Q-type channels play key roles in regulating spike firing properties and contributing to Ca^2+^ transient/complex spikes that result from climbing fiber activity [36]. Moreover, they regulate heterosynaptic competition between climbing fibers and parallel fibers and also drive homosynaptic competition among multiple climbing fibers [37]. In contrast to human *CACNA1A* dominant mutations, the first animal model to be characterized showed predominantly recessive mutation in the *CACNA1A* gene (tottering mice) [38,39]. The *tottering* (*tg*) mice display mildly ataxic behavior during locomotion [40,41,42]. In *tg* Purkinje cells, the P/Q-type current density is decreased by ~40% [43] and spike firing patterns show enhanced irregularities with periods of pauses and bursts [41]. Consistent with a reduced functional role of P/Q-type channels, parallel fiber–Purkinje cell synapses are impaired in *tg* mutants [44]. Additionally, electron microscopic and Golgi-staining procedures have revealed shrunken Purkinje cells with a reduced size in the soma, abnormal Purkinje cell connectivity and diffuse axonal swellings [45,46,47]. Two additional recessive *tg* mutations have been identified showing different but overlapping features (*leaner* and *rolling Nagoya* mice). As opposed to *tg*, *leaner* (*tg^la^*) and *rolling Nagoya* (*tg^rol^*) mice are severely ataxic [39,48,49]. In addition to causing a reduction in P/Q-type current density by ~60%, the *tg^la^* mutation affects Ca^2+^ channel gating kinetics [43]. In detail, Ca^2+^ channel currents in *tg^la^* Purkinje cells show a distinct change in the voltage dependence of activation and inactivation. Furthermore, these mice exhibit Purkinje cell degeneration whose parasagittal striped pattern is similar to the pattern of zebrin expression [39,50]. Likewise, electrophysiological studies have demonstrated that also the *tg^rol^* mutation in Purkinje cells results in the reduced voltage sensitivity (i.e mutated channels are less sensitive to voltage stimuli) and diminished activity of P/Q-type channels (~40%) [51,52]. Overall, morphological investigations have revealed characteristic synaptic alteration between parallel fiber varicosity and Purkinje cell dendritic spines in all the three mutant mouse models of SCA. Multiple Purkinje cell dendritic spines synapse with single parallel fiber varicosity [47]. A recently described ataxic model in rats (*GRY*; groggy) is also featured by a recessive mutation [53]. The phenotype of the *GRY* rat resembles that of the *tg* mouse rather than that of the other two mutant mice. In 2007, however, Xie and colleagues reported the first dominant ataxic mouse model of *CACNA1A* mutation, called *Wobbly* (*wb*) on the basis of its peculiar gait [54]. Heterozygotes show severe ataxia and reduced locomotor activity with associated degeneration of Purkinje cells [54,55]. Purkinje cells display altered passive and active membrane properties and an altered synaptic excitation/inhibition balance in *wb* mice. Specifically, *wb* Purkinje cells are less excitable showing increased resting membrane potential and action potential threshold. Parallel fiber stimulation fails to evoke excitatory synaptic currents in more than 50% of Purkinje cells, while evoked synaptic inhibition is shown to be stronger [55]. Later, another dominant mutation, known as *Tg-5J*, in *CACNA1A* gene, resembling the *wobby* and many human mutations, was described by Miki and coworkers [56]. Heterozygotes *Tg-5J* mice are extremely ataxic and homozygotes rarely survive. The *Tg-5J* mutation determines a negative shift in the P/Q-type channel activation curve despite of no significant changes in the Ca^2+^ current density [56]. Moreover, the dendritic arbor is significantly reduced in complexity and overall mass in *Tg-5J* Purkinje cells [56]. Since SCA6 share features in common with other polyglutamine (poliQ) diseases, it is reasonable to expect that the poliQ expansions would affect the P/Q-type channel function. Different studies, performed on several lines of knockin (KI) mice carrying expanded or hyperexpanded CAG repeat tracts in the *CACNA1A* gene, have achieved highly variable, often conflicting results. Matsuyama and colleagues showed that a P/Q-type channel with 24 polyglutamines (*SCA624Q*) exhibited normal gating properties whereas 40 expanded polyglutamines (*SCA640Q*) caused an hyperpolarizing shift in the voltage dependence of inactivation [57], suggesting an overall reduction of Ca^2+^ influx. On the contrary, another study revealed that showed the voltage dependence of inactivation of P/Q-type channels was negatively shifted in *SCA624Q* mice [58]. Recently, KI mice were generated by introducing hyperexpanded polyglutamines (*SCA684Q*) [59,60,61,62]. These mice develop progressive motor impairment in adult age. Extracellular recordings of cerebellar slices from *SCA684Q* mice revealed a decrease of Purkinje cell firing (<100 Hz) [63] accompanied by a reduction of the spike timing precision in spite of unchanged P/Q-type channel kinetics and current density, suggesting that expanded CAG repeat per se does not affect the intrinsic properties of the channels. In addition, surplus climbing fiber inputs on developing *SCA684Q* Purkinje cells were observed [63]. The aforementioned mouse models with intrinsic properties affected in P/Q-type channels, such as pore conduction, gating and inactivation kinetics, can provide further opportunities to understand how perturbations in Ca^2+^ signaling regulate neuronal excitation and its underlying relation with human SCAs.

## 3. Mutations in the *CACNA1G* Gene

SCA42 is a rare, autosomal dominant neurological disorder characterized by pure and slowly progressive cerebellar defects, including incoordination of gait, dysarthria, nystagmus, and saccadic eye movements (Table 1). Post-mortem histological examination revealed cerebellar atrophy in association with a prominent Purkinje cell loss [64]. A heterozygous single point mutation in the *CACNA1G* gene which causes an arginine-to-histidine (*p.Arg1715His*) change in the S4 voltage-sensing region of the T-type voltage-gated Ca^2+^ channel protein Ca(v)3.1 has concurrently been associated to SCA42 in ten families [64,65,66,67]. T-type Ca^2+^ channels are widely expressed in the brain, including the thalamus, hippocampus, neocortex, and cerebellum [68]. In situ hybridization showed that Ca(v)3.1 is the major isoform of T-type Ca^2+^ channels expressed in cerebellar motor circuit, including Purkinje cells, DCN and inferior olivary (IO) neurons, a result that is also supported by immunocytochemical localization [68,69,70,71]. The unique electrophysiological properties of T-type Ca^2+^ channels (especially the low voltage-activated Ca^2+^ current) are well suited to regulating neuronal excitability and oscillatory behavior near the resting membrane potential [72]. Specifically, T-type Ca^2+^ channels have a significant role in dendritic Ca^2+^ spikes and the resultant spontaneous bursts in Purkinje cells [73,74], whereas they mediate a rebound burst firing after hyperpolarization in DCN neurons [71] and underlie resonance and subthreshold membrane potential oscillations in IO neurons [75,76]. Recent findings on SCA42 using whole-exosome sequencing in HEK293T cells found that arginine-to-histidine change in *CACNA1G* gene was able to determine a positive shift of T-type Ca^2+^ channel voltage-dependence [65]. In order to elucidate whether this point mutation is implicated in the SCA42 phenotype, a new transgenic (*Cacna1g-Arg1723His*) mouse model, harboring the same mutation identified in the SCA42 families, has been generated [77]. *Cacna1g-Arg1723His* mice showed an adult-onset ataxic phenotype concomitant with cerebellar atrophy and Purkinje loss, as reported in human SCA42. Furthermore, *Cacna1g-Arg1723His* Purkinje cells displayed a change in the voltage-dependence of T-type Ca^2+^ channels which strongly impact on their excitability [77]. Interestingly, Hashiguchi’s study also described that *Arg1723His* mutation altered the resonance properties of IO neurons, a result that has also been reported in Ca_v_3.1 KO mice [78,79]. Since Ca_v_3.1 KO mice are reported to have normal growth, normal brain weight and structure, and no significant defects in motor learning and motor coordination were observed [80,81], the *CACNA1G* gene mutation most likely causes ataxic symptoms through a process different from a classic loss-of-function mechanism.

## 4. Mutations in the *GRID2* Gene

Recently, *GRID2* point mutations with putative gain-of-function mechanisms were reported in several patients diagnosed with dominant inheritance of cerebellar ataxia [82]. They affect the same amino acid as that previously described by Phillips in *Lurcher* mouse model [83]. *GRID2* gene encodes for the δ2 glutamate (GluD2) receptor, which is mostly expressed in Purkinje cells, where it is co-localized with AMPA receptors at the postsynaptic density (PSD) of parallel fiber synapses [84,85]. Although GluD2 receptors belong to the ionotropic glutamate receptor family [86], they are not gated by L-glutamate but selectively bind to D-serine and glycine [87]. GluD2 receptors are required for proper development and function of the cerebellum [88,89]. They play a crucial role in Purkinje cell synapse formation by interacting with presynaptic proteins [87,90]. Actually, several studies have demonstrated motor incoordination related to ataxia in GluD2 receptor KO mice. In addition, the number of contacts between parallel fiber and Purkinje cell is reduced while the innervation by climbing fibers into the distal part of the Purkinje cell dendritic tree is extended, thus invading the parallel fiber territory [89,91,92,93]. Furthermore, LTD impairment was observed at the parallel fiber–Purkinje cell synapse [89]. Homozygous *Lurcher* mice die at birth [94], whereas heterozygotes exhibit prominent ataxic gait and motor coordination deficit as a result of selective cell-autonomous apoptosis of Purkinje cells during the third postnatal week, when about 90% of the Purkinje cells have disappeared [95,96,97]. *Lurcher* Purkinje cells have a very high membrane conductance and a depolarized resting potential due to a large inward current resulting from continuous Na^+^ and Ca^2+^ influx through constitutively open GluD2 receptors in the absence of ligand binding [6,97,98,99]. The mechanism by which continuous ion flow through mutated receptors is capable of inducing Purkinje cell death was proposed by [100] (Figure 1).

In response to the constitutive ion influx through *Lurcher* GluD2 receptors, intracellular ATP levels decrease probably by overactivation of the Na^+^/K^+^-ATPase. Indeed, in *Lurcher* Purkinje cell dendrites, the mitochondrial oxidative respiration is increased in order to balance the energy demands of removing excess Na^+^ and Ca^2+^ entering the cells in response to the constitutive GluD2 current [96]. Compromised ionic homeostasis together with decreased ATP levels can lead to cell swelling and subsequent cell death. Furthermore, Ca^2+^ influx through VGCCs could activate a variety of Ca^2+^-dependent enzymes, such as the cysteine protease calpain, and potentially contribute to Purkinje cell death through different pathways. Studies to assess the contribution of intracellular Ca^2+^ levels in triggering downstream pathways mediating Purkinje cell death in *Lurcher* mice will be directly relevant for our understanding the mechanisms of neurodegenerative diseases, such as SCAs. In conclusion, *Lurcher* mice will continue to be a valuable model for the development of disease-targeted therapeutic approaches.

## 5. Mutations in the *PRKCG* Gene

Spinocerebellar ataxia type 14 (SCA14; Table 1), characterized by slowly progressive cerebellar dysfunction, dysarthria and abnormal eye movements, is caused by almost 40 different mutations in *PRKCG* gene, encoding the protein kinase C gamma (PKCγ) [19,101]. PKCγ is a member of the classical PKC subfamily which is activated by Ca^2+^ and the second messenger diacylglycerol (DAG), which is synthesized upon phosphatidylinositol 4,5-bisphosphate (PIP_2_) hydrolysis by phospholipase C (PLC) [102]. Whereas an increase in intracellular Ca^2+^ concentration ([Ca^2+^]_i_) is required to stimulate the cytosolic PKCγ to translocate towards the plasma membrane, subsequent DAG binding is mandatory for enzyme activation. PKCγ is expressed solely in the brain and spinal cord and its localization is restricted to neurons [103]. In the cerebellum, PKCγ is most abundant in Purkinje cell soma and dendritic processes where it plays a role in normal development of the climbing fiber input from the inferior olive [104,105,106,107]. In both PKCγ KO mice and mice expressing a transgenic PKCγ inhibitor, multiple climbing fiber innervation of Purkinje cells persists into adulthood [106,108,109]. Nonetheless, the PKCγ KO mice only show mild ataxia and no gross morphological abnormalities in Purkinje cells [106]. In contrast, cerebellar atrophy and loss of Purkinje cells were described at post-mortem in SCA14 patients [110,111]. These studies show the importance of PKCγ activity in Purkinje cell development and neuronal connectivity and suggest the possibility that a gain of toxic function rather a loss of function of PKCγ underlies the pathogenic mechanism of SCA14. A previous report showed that two SCA14 mutations led to increased PKCγ intrinsic activity. The mutant kinase displays enhanced Ca^2+^-induced membrane translocation, which links Purkinje cell loss to a potential gain of PKCγ-mediated signal transduction [112]. More recently, Adachi and co-workers [113] have studied the PKCγ activity in 20 spontaneous mutations found in *PRKCG* gene of SCA14 patients. Nineteen of the 20 mutations showed an increased PKCγ activity although they were unable to bind DAG and their residency time within the plasma membrane was significantly shorter. As a consequence, the mutant PKCγ could not phosphorylate and inhibit extracellular Ca^2+^ entry through Transient Receptor Potential Canonical 3 (TRPC3). Of note, TRPC3 provides a second messenger-operated Ca^2+^-entry pathway which is gated by DAG upon stimulation of metabotropic receptors [114]. As a consequence, [Ca^2+^]_i_ is remarkably increased in the presence of mutant PKCγ, thereby leading to aberrant intracellular signaling. These results indicate that an alteration in Ca^2+^ homeostasis and Ca^2+^-mediated signaling in Purkinje cells may be responsible for the neurodegeneration characteristic of SCA14. However, another report showed that SCA14 mutant PKCγ activity is reduced in living cells causing an inefficient activation of downstream mitogen-activated protein kinase (MAPK) signaling. These events might therefore lead to reduced expression of target genes and affect neural development and survival [115,116]. To date, only two transgenic mouse models of SCA14 have been generated: PKCγ-H101Y and PKCγ-S361G mice, respectively [117,118]. Although the PKCγ-H101Y transgenic mouse express H101Y mutation under a universal promoter, PKCγ-H101Y is peculiarly expressed in Purkinje cell soma and dendrites. The authors report an altered morphology and loss of Purkinje cells at the age of four weeks combined with an ataxic phenotype which is more severe than PKCγ KO mice [117]. In the most recent transgenic mouse model of SCA14, by using the L7 promoter, the S361G mutation of the PKCγ was specifically expressed in Purkinje cells [118]. Purkinje cells from PKCγ-S361G transgenic mice show severe inhibition of dendritic development which is identical to Purkinje cells treated with a PKCγ agonist, motor deficits typical for cerebellar ataxias and Purkinje cell degeneration and loss in a localized area (lobule 7). These findings support the notion that the increased activity of PKCγ results in the functional cerebellar deficit usually recognized in SCA14 patients. Interestingly, point mutations inducing other forms of SCA often affect signaling proteins which are involved in the PKCγ pathways, such as TRPC channels and inositol-1,4,5-trisphophoshate receptors (IP_3_Rs). Since PKCγ is likely to play a crucial role for proper Purkinje cell function, it might be an appealing drug target for the treatment of SCAs [118].

## 6. Mutations in the *TRPC3* Gene

The *Moonwalker* mouse (*Mkw*) is a recently described model of dominantly inherited cerebellar ataxia that displays impaired motor and coordination control associated with a slow but progressive loss of Purkinje cells [119,120]. The phenotype is caused by a gain of function mutation (T365A) of the *TRPC3* gene encoding for the Ca^2+^-permeable, non-selective cation channel TRPC3 (Transient Receptor Potential Canonical 3) [119,120,121]. TRPC3 mediates extracellular Ca^2+^ entry in response to G_q/11_-Protein Coupled Receptor (GPCRs) or Tyrosine Kinase Receptors (TKRs), thereby modulating a host of several physiological functions, including synaptic transmission, arterial tone regulation and angiogenesis [122,123,124,125,126,127]. TRPC3 channels are abundantly expressed in the cerebellum, particularly in Purkinje cells and unipolar brush cells (UBCs), a small glutamatergic neurons mainly found in the granular layer of vestibulocerebellum [119,128,129]. During Purkinje cell postnatal development, TRPC3 channel expression is upregulated promoting a rapid dendritic growth and synapse formation [130]. Electrophysiological recordings from *Trpc3* KO Purkinje cell have established that TRPC3 channels mediate slow postsynaptic currents (sEPSCs) in response to stimulation of subtype I metabotropic glutamate receptors (mGluR1) [123,129]. When stimulated, mGluR1 activates PLCβ to hydrolize PIP_2_ into IP_3_ and DAG [131]. As a consequence of IP_3_-induced endoplasmic reticulum (ER)-Ca^2+^ store depletion, the ER-Ca^2+^ sensor Stromal Interaction Molecule 1 (STIM1) translocates to peripheral ER cisternae, where it aggregates within sub-membranal clusters and gates Orai1 [132] and/or TRPC3 channels [133,134,135,136]. In addition to mediating sEPSCs (Hartmann et al., 2008), the functional coupling between mGluR1 and TRPC3 channels was found to be critical for the induction of long-term depression (LTD) at the parallel fiber–Purkinje cell synapse [124,125]. Finally, a recent study has revealed that the difference in intrinsic firing activity of zebrin II-positive and zebrin II-negative Purkinje cells is generated by differential activity of TRPC3 channels [137]. In the *Mkw* mutation, threonine 365 is converted into alanine in the highly conserved S4/S5 linker region of the TRPC3 protein leading to abnormal channel gating [119,128]. Accordingly, TRPC3 was activated following low mGluR1 stimulation in *Mkw* Purkinje cells (Becker et al., 2009). In addition, these cells are reduced in size and show a lower complexity of dendritic arborization which is characterized by few and short branches. Concurrent with dendritic abnormalities, a significant decrease in climbing fiber synapses along dendrites is observed [119,120]. UBCs are also severely reduced contributing to pronounced balance impairment of *Mkw* mice [128]. As Ca^2+^ signaling plays a fundamental role in regulating dendritogenesis and connectivity in the developing brain [138] and *Mkw* phenotype derives from a gain-of-function in the Ca^2+^-permeable TRPC3 channel, an abnormal Ca^2+^ influx during the critical period of development has been suggested to be the cause of limited dendritic arborization in *Mkw* Purkinje cells. Based upon the interaction between PKCγ and TRPC3 described above, Becker and colleagues [119,121,125] have proposed a model in which mGluR1-TRPC3-PKCγ signaling at parallel fiber–Purkinje cell synapse is essential in controlling developmental and functional processes [129,130,139,140] (Figure 2).

TRPC3 channels are present in the same protein complex with mGluR1and both receptors localize to Purkinje cell soma and dendrites [119,130,140]. Moreover, TRPC3 channels and PKCγ primarily respond to PLC-coupled receptors, such as mGluR1. Finally, TRPC3 channel activity has been shown to be negatively regulated through phosphorylation by PKCγ at the distinct site of threonine 365, as also discussed above [113,119,141,142]. Thus, the absence of the PKCγ-mediated negative feedback on TRPC3 channels, which could occur upon mutations in the *PRKCG* gene, might determine a prolonged activation, resulting in sustained high levels of intracellular Ca^2+^ into Purkinje cells. Of note, besides modulating TRPC3 activity via PKCγ, DAG is also able to gate TRPC3 in a non-PKC-dependent manner [122]. Thus PKC is a powerful inhibitor of TRPC3 channels while DAG is a dual regulator [136]. In addition, TRPC3-mediated sEPSCs may be also induced following mGluR1-dependent activation of phospholipase D (PLD) through the small GTP-binding protein Rho [143]. In addition, recent studies have demonstrated that mGluR1 activation may be sustained by the physical association with the GluD2 receptors. GluD2 receptors share remarkable sequence homology with ionotropic glutamate receptors, although they neither bind to glutamate nor conduct ion currents [144]. A recent investigation demonstrated that mGluR1 may physically associate with GluD2, PKCγ, and TRPC3 in mouse cerebellar Purkinje neurons, thereby favoring the cell surface expression of the protein complex and boosting the efficacy of mGluR1-mediated synaptic transmission [145,146]. Perturbed TRPC3 signaling is likely to be a common pathological mechanisms in other genetic forms of cerebellar ataxia. Indeed, *TRPC3* gene was recently identified as a central player in mouse models of SCA14 [113,129], SCA1 [147,148] and SCA2 [149] (see paragraphs above). SCA1 and SCA2 are caused by an abnormally expanded CAG-repeat sequence in the Ataxin-1 (*ATXN1*) and Ataxin-2 (*ATXN2*) genes, respectively. In transgenic mice expressing the human *ATXN1* and *ATXN2* genes with an expanded CAG tract under the Purkinje cell-specific *Pcp2* promoter, *TRPC3* gene was downregulated at an early stage in pathogenesis [148,149]. Intriguingly, the first functionally pathogenic variant in the human *TRPC3* gene was identified in a patient with adult-onset cerebellar ataxia (SCA41; Table 1) [150,151]. Collectively, these findings point to a crucial role of TRPC3 signaling in hereditary forms of human cerebellar ataxia, making the *TRPC3* gene a promising candidate for screening ataxic patients with unknown genetic etiology [152].

## 7. Mutations in the *Itpr1* Gene

Inositol 1,4,5-trisphosphate receptors (IP_3_Rs) are intracellular IP_3_-gated channels that release Ca^2+^ from the endoplasmic reticulum (ER). IP_3_ is generated from PIP_2_ hydrolysis by PLCγ and PLCβ in response to various extracellular stimuli acting on GPCRs and TKRs [153]. In mammals, including humans, the IP_3_R family consists of three isoforms (types 1–3) encoded by distinct genes (*Itpr1*, Itpr2, and Itpr3). Although human IP_3_Rs share around 70% sequence homology, they exhibit a different sensitivity to IP_3_ and tissue-/cell specific distribution [154,155,156,157]. IP_3_ receptor type 1 (IP_3_R1) is primarily expressed in the cerebellum, especially in Purkinje cells, but it is also localized in other brain areas, including the cerebral cortex, basal ganglia, thalamus, and hippocampus [154,158,159,160]. Several reports suggest that abnormal IP_3_-mediated Ca^2+^ signaling is involved in the pathogenesis of spinocerebellar ataxia types 15/16 (SCA F; Table 1) and 29 (SCA29; Table 1), as well as other neurodegenerative disorders, such as Alzheimer’s disease (AD) and Huntington’s disease [161,162]. Van de Leemput and colleagues first described a heterozygous deletion of *Itpr1* gene, encompassing exons 1–10, 1–40, and 1–44 in Australian and British families of patients for SCA15, which is characterized by the adult-onset of very slowly progressive gait and limb ataxia with episodic dystonic symptoms [163,164,165,166,167]. Soon after, a complete deletion of the gene was found in a Japanese patient with SCA15 [168]. A subsequent study reported only a heterozygous deletion limited to exons 1-48 of the *Itpr1* gene in a patient with SCA16 [169], indicating that haploinsufficiency of *Itpr1* gene was the cause of both SCA15 and SCA16. Gardner proposed to designate SCA16 as a “vacant SCA”; however, the term SCA15/16 is still widely employed [170]. At least six different heterozygous point mutations have been identified in SCA29 which is distinguished by infantile-onset (congenital) motor development delay followed by non-progressive ataxia and mild cognitive impairment [171,172]. One of this mutation is located in the allosteric inhibitor of IP_3_R1, the carbonic anhydrase VIII-binding protein (Car8; [173]). There are two spontaneous mutant mice that naturally possess an *Itpr1*gene mutation, *ITPR1^opt/opt^* mice and *ITPR1^Δ18/^^Δ18^* mice. Both the mouse models show a decrease in the normally high level of IP_3_R1 expression in the cerebellum [164,174]. Their phenotype start to show ataxic and convulsive phenotypes around 10 days after birth and die within 4 weeks [164,175]. In the regulatory domain of the *Itpr1* gene*, ITPR1^opt/opt^* and *ITPR1^Δ18/^^Δ18^* mice have a homologous in-frame deletion of exons 43/44 and exon 36, respectively [164]. As a result of this deletion, several putative modulatory sites are removed [154,175,176]. Interestingly, the *opt*-IP_3_R1 seems to be more subject to proteolysis than the wild-type IP_3_R1 isoforms. Additionally, the single-channel conductance of the *opt*-IP_3_R1 is reduced by 20% and sensitivity to potentiation by ATP is reduced 20-fold [177]. Surprisingly, Street and colleagues [175] reported that, despite markedly decreased IP_3_R1 protein level, repetitive mGluR stimulation elicited a supranormal Ca^2+^ release from ER in *ITPR1^opt/opt^* Purkinje cells. These data suggest that the phenotype observed in *ITPR1^opt/opt^* mice may be caused by the physiological misregulation of a functional IP_3_R1. Currently, Ca^2+^ measurements and membrane electrophysiology would be necessary in order to ascertain whether *ITPR1^Δ18/^^Δ18^* mice match the physiology of SCA15/16. Lastly, IP_3_R1 KO mice show a postnatal phenotype similar to those of *ITPR1^opt/opt^* and *ITPR1^Δ18/^^Δ18^* mice [178,179]. Purkinje cells from IP_3_R1 KO mice exhibit abnormal dendritic arborization and enlarged parallel fiber terminals characterized by an accumulation of synaptic vesicles (Hisatsune et al., 2006). Hisatsune and colleagues suggested that BDNF production through IP_3_R1-mediated Ca^2+^ signaling in cerebellar granule cells intercellularly controls the dendritic outgrowth of Purkinje cells, resulting in altered parallel fiber–Purkinje cell synaptic efficacy. Indeed, electrophysiological experiments in IP_3_R1 KO mice demonstrated that long-term depression (LTD) at the parallel fiber–Purkinje cell synapse was impaired [180]. Furthermore, recordings in freely behaving conditional cerebellum/brainstem IP_3_R1 KO mice revealed distinct patterns of Purkinje cell firing linked to the dystonic symptoms which could be rescued by the deletion of Purkinje cells or by pharmacological inactivation of the inferior olive [181]. Taken together, the spectrum of *ITPR1*-associated phenotypes is intriguing. The heterozygous *Itpr1*gene deletions in late onset SCA15 suggest haploinsufficiency as a disease mechanism. Conversely, cases with a congenital or infantile onset of SCA29 appear to be caused exclusively by *Itpr1* gene point mutation that alter the structure of IP_3_R1, presumably through a dominant negative effect [182].

## 8. Mutations in the *ATXN2/ATXN3* Genes

SCA2 and SCA3 are two of the more prevalent polyQ diseases caused by CAG repeat expansion in the genes encoding ataxin-2 (*Atxn2*) and ataxin-3 (*Atxn3*) proteins which are widely expressed in neuronal tissue [183,184,185,186,187,188,189]. Typically, mutant proteins have an increased tendency to aggregate forming ubiquitinated microscopically visible intranuclear inclusions [189]. SCA2 is characterized by progressive cerebellar ataxia, dysarthria, and oculomotor deficits [190,191] whereas SCA3, also known as Machado–Joseph disease (MJD), is a multi-systemic disorder affecting various different regions of the brain and spinal cord. Although the cerebellar cortex and olivary nuclei are relatively spared, high-resolution magnetic resonance imaging shows cerebellar atrophy. SCA3 is characterized by a broad spectrum of clinical signs involving cerebellar, pyramidal, extrapyramidal and spinal motor functions [192,193]. The role of Ca^2+^ signaling in the pathogenesis of SCA2 is supported by the genetic inverse correlation between CAG-repeat length in the *CACNA1A* gene and the age of disease onset [194] and further reinforced by finding that mutant *Atxn2* has been shown to interact with the cytosolic COOH-terminal of IP_3_R1 by increasing the sensitivity of the receptor to its ligand [20]. Transgenic animals expressing 58 glutamine repeats in the *Atxn2* gene under the control of the L7 promoter (SCA2-58Q) exhibit Purkinje cell loss or marked changes in the Purkinje cell dendritic arborization, without nuclear localization or a detectable increase in ubiquitin-conjugated protein complexes [189]. These features are typical for SCA2 patient pathology. Moreover, specific behavioral assays quantified by rotarod and beam-walk analysis reveal age-dependent motor coordination deficits [20]. These reports are consistent with a gain-of-function or toxic gain-of-function of mutant *Atxn*2. In addition, mice deficient in *Atxn2* do not show any evidence of ataxia or major morphological abnormalities in the cerebellar cortex [195]. This evidence further supports the notion that CAG repeat expansion in *Atxn2* does not cause a loss of function nor does it have a dominant negative properties. Recent analyses of Purkinje cell pacemaking activity recorded from SCA2-58Q transgenic mice using extracellular single-unit in vivo recordings demonstrated an age-dependent bursting and irregular firing patterns [196,197]. Furthermore, Ca^2+^ imaging recordings showed a significant increase in Ca^2+^ release from ER via IP_3_R1 upon mGluR stimulation resulting in significantly more cell death [20]. Chronic suppression of IP_3_R-mediated Ca^2+^ signaling by the over-expression of IP_3_ 5-phosphatase (5PP) using adeno-associated virus in aging SCA2-58Q transgenic mice has been shown to prevent Purkinje cell death, normalize their firing pattern, and attenuate motor coordination deficits [197]. The mechanism by which the IP_3_R1-mutant *Atxn*2 interaction results in increased IP_3_R1 activity leading to deranged neuronal Ca^2+^ signaling was proposed by Kasumu and colleagues [22]. Aberrant IP_3_R1-dependent ER Ca^2+^ release results in the mitochondrial Ca^2+^ overload, release of cytochrome C and, consequently, induction of Purkinje cell death via dark cell degeneration (Figure 3), which represents a peculiar mode of excitotoxic cell death.

Interestingly, the positive modulation of small conductance Ca^2+^-gated K^+^ channels type 2 (SK2), which are known to play an important role in the regulation of Purkinje cell spiking activity [198,199], restores regular firing pattern of 58-Q Purkinje cells [200]. Furthermore, long-term feeding of aging SCA2-58Q transgenic mice with a novel selective positive modulator of SK2 channels (NS13001) improves motor performance [200]. This result identifies NS13001 as a potential therapeutic drug for treatment of SCA2. It is conventionally presumed that, except for *Atxn2* which has an exclusively cytoplasmic localization in normal and SCA2 human brain [201], nuclear expression of CAG-expanded proteins is required for the pathogenesis of polyQ diseases [11,202,203]. Recently, genetic experiments, using transgenic mice expressing CAG-expanded *Atxn3*-79Q (SCA3-79Q), which display various symptoms of motor dysfunction, have confirmed the nuclear toxicity of mutant *Atxn3* (Bichelmeier et al., 2007). In addition to forming toxic aggregates, single-channel recordings in reconstituted lipid bilayer membranes revealed that mutant *Atxn3* may increase IP_3_R activation by IP_3_, thus destabilizing neuronal Ca^2+^ signaling [161,204]. However, blocking the Ca^2+^-dependent proteiase, calpain, abrogated the formation of inclusions in cells expressing the mutant *Atxn3* [205]. Moreover, in SCA3-79Q transgenic mice, pharmacological inhibition of Ca^2+^-induced Ca^2+^ release through ryanodine receptors (RyRs) was shown to counterbalance the increased activity of IP_3_R1, thereby resulting in an improvement of their performance in motor coordination assays [204]. These results strongly support the role of deranged Ca^2+^ signaling in the pathogenesis of SCA3. Finally, in SCA3-79Q transgenic mice, microarrays analysis and RT-PCR assays indicated downregulated expression of cerebellar genes involved in signal trasduction, neurotransmission or synaptic plasticity, including those encoding IP_3_R1, PLCγ and calcineurin [206]. Whole-cell patch-clamp recordings provided the evidence that mutant *Atxn3*-induced transcriptional repression impaired LTD induction at the parallel fiber–Purkinje cell synapse in SCA3-79Q mice [206]. These findings are at odds with the reported gain of function of IP_3_R1 following binding to the mutant *Atxn3* [161,204]. These contradictory findings suggest that it could not be the reduced or increased activity of IP_3_R1 but rather the loss of a precise regulation of its activity that triggers the disease.

## 9. Mutations in the *ATXN1* Gene

Expansion of CAG repeats within the *ATXN1* gene encoded the cytosol/nuclear protein ataxin-1 (*Atxn1*) causes the adult-onset neurodegenerative disease SCA1 characterized by progressive ataxia, oculomotor deficits, pyramidal/extrapyramidal signs [207,208,209,210,211]. *Atxn1* is widely expressed in the normal human brain and peripheral tissues. Immunohistochemical studies have revealed that *Atxn1* is predominantly nuclear in neuronal cell types although Purkinje cells also present a minor cytoplasmic component [212]. The primary neurodegeneration in SCA1-affected individuals results in Purkinje cell loss and atrophy of specific brainstem neurons [208,213,214,215,216]. Moreover, nuclear inclusions consisting of the mutant *Atxn1* have been described in neurons of SCA1 patients [217,218]. Interestingly, by using RNA in situ hybridization, high levels of *Atxn1* expression in murine Purkinjie cells were detected during postnatal maturation when the growth of the dendritic tree is being completed and synaptic connections are being completed and becoming functional. This finding suggests that *Atxn1* might play a role in Purkinje cell development and function [219,220]. Because the normal form of *Atxn1* as well as *Atxn1* with the expanded CAG repeat are transcribed and translated in the tissues of affected SCA1 patients, and the absence of *Atxn1* in the mouse does not lead to any ataxic symptoms or major motor coordination abnormalities [221], the toxic-gain of function exerted by mutant *Atxn1* has been suggested. In order to test this hypothesis, different SCA1 transgenic mice have been generated. The first transgenic mouse model of SCA1 employed the Purkinje cell-specific promoter L7 to precisely target the expression of the human *ATXN1* gene [222]. Two lines were created with various repeat lengths; a control line with over-expression of the normal allele containing 30 CAG repeats (30Q), while the expansion construct contained 82 CAG repeats (82Q). *ATXN1*(30Q) mice exhibit a normal phenotype with no signs of neurological disorders. Conversely, *ATXN1*(82Q) mice reveal severe ataxia and progressive Purkinje cell loss [222,223,224]. *ATXN1*(82Q) Purkinje cells develop two pathological features described in cerebellar tissue from patients with SCA1: atrophic dendritic arborization and nuclear aggregates of mutant *Atxn1* [224]. Using Ca^2+^-imaging and electrophysiological approaches in *ATXN1*(82Q) mice, a recent study reported a significant reduction in the responsiveness of Purkinje cells to climbing fiber activation at early-disease stage (~6 weeks) while parallel fiber–Purkinje cell transmission alteration occurred during SCA1 progression (28–40 weeks) [225]. Furthermore, immunofluorescent labeling of climbing fiber terminals revealed a reduction of climbing fiber synapses in the distal segment of the Purkinje cell dendritic tree resulting in incomplete innervation territory [225]. Finally, several specific genes involved in Ca^2+^ homeostasis have been shown to be sequentially downregulated in *ATXN1*(82Q) Purkinje cells, with IP_3_R1 and the Sarco-Endoplasmic Reticulum Ca^2+^-ATPase 2 (SERCA2) affected first, followed by TRPC3 channels [148]. A similar downregulation has also been found in SCA1 human cerebellum. Moreover, changes in the dendritic expression of the Ca^2+^-binding proteins parvalbumin (PV) and calbindin D-28k (CaB) were observed [223,226]. Lastly, in order to assess the role of PV and CaB in the pathogenesis of SCA1, double mutant animals were generated by crossing CaB KO with *ATXN1*(82Q) mice, leading to an increased disease phenotype [227]. Although indirectly, these findings strengthen the suggestion that the downregulation of proteins implicated in Ca^2+^ homeostasis and Ca^2+^ signaling in Purkinje cells can be an important component in the etiology of SCA1. Further understanding of Ca^2+^ signaling mechanisms could offer clues to the design of novel and effective therapeutic strategies for treatment of spinocerebellar ataxias.

## 10. Mutations in the *GMR1* Gene

mGluR1, encoded by *GRM1* gene, is one of the most abundant of its group of receptors in the mammalian central nervous system and is expressed at particularly high levels in Purkinje cells. Disease caused by mutations in *GRM1* gene are extremely rare [228]. To date, the only *GRM1* mutations identified have been found to cause an autosomal recessive spinocerebellar ataxia (SCAR13; [229]). Recently, Watson and colleagues reported heterozygous dominant mutations in *GRM1* gene that are associated with distinct disease phenotypes: gain-of-function point mutations that lead to enhanced receptor activity causing an adult-onset cerebellar ataxia and a truncation mutation that result in a dominant-negative effect causing a juvenile-onset cerebellar ataxia characterized by cognitive impairment (SCA44; Table 1) [230]. mGluR1 loss of function has been revealed in some animal models of human cerebellar ataxia, such as SCA1 and SCA3 transgenic mice. In *ATXN1*(82Q) mice, it has been shown that the progressive loss of mGluR1 function is restored by the GABA_B_ receptor agonist baclofen leading to rescue the motor coordination [223,231,232,233]. In SCA3 Purkinje cells, mGluR1 have been found to be mislocalized to non-synaptic site resulting in a disruption of mGluR1 signaling [234]. On other hand, mutations that causes a mGluR1 gain of function have been associated with SCA. For example, in SCA2-127Q transgenic mice, the significant increase in Ca^2+^ release from ER via IP_3_R1 caused by an amplification of mGluR1 signaling is prevented by buffering basal Ca^2+^ concentration at normal resting levels [20,235,236]. In conclusion, functional alteration of mGluR1 activity has been documented in different SCA mouse model, while, in human patients, several forms of cerebellar ataxia result from mutations in genes in this pathway, which suggests that disruption of mGluR1 signaling and downstream Ca^2+^ homeostasis form a common pathological mechanism underlying SCAs [21].

## 11. Prospects for Therapeutic Development

Currently, there is no preventive or curative treatment for SCAs, but over the last few years, several drugs and therapeutic strategies that potentially slow the disease progression are being tested. The therapeutic approaches can be divided into pharmacological and gene therapies that target the toxic downstream effect and stem cell replacement (neurotransplantation). Since disrupted Purkinje cell Ca^2+^ signaling has been identified as a common underlying mechanism in most of SCAs (Table 2), one pharmacological approach has been to propose that Ca^2+^ blockers and stabilizers have a potential utility for patients diagnosed with SCAs.

Long-term feeding with intracellular dantrolene (Ca^2+^ stabilizer) resulted in improved motor performance and decreased Purkinje cell loss in mouse models of SCA2 and SCA3 [20,204,237]. Similar effects were demonstrated for a novel Itpr1 inhibitor T-558 [238]. Another interesting pharmacological approach, with some preclinical success, has been the use of caffeine. In a mouse model of SCA3 administration of caffeine through the drinking water alleviated motor deficits [239]. Finally, the treatment with glutamate release inhibitor, riluzole, in a randomized placebo-controlled clinical trial in 40 SCA patients for 12 months (SCA1, 2, SCA6), improved ataxic symptoms [240]. Gene therapy has made great progress over the past decade, especially in the cases of the polyglutamine (polyQ)-repeat SCAs. Currently, the most encouraging and innovative are RNA-targeting therapies including RNA interference (RNAi) or antisense oligonucleotides (AONs), which have been validated in different mouse models [241]. RNAi is an evolutionarily process of post-transcriptional gene suppression initiated by double-stranded RNA sequences to reduce mRNA expression. RNAi can be induced by microRNAs (miRNA) or small hairpin RNA (shRNA). By causing a severe degradation of mRNA, miRNAs and shRNA are considered potent tools to achieve gene silencing. A recent study found that miRNA expression in the human cerebellum was downregulated in aging brain resulting in an increase of ATXN1 gene expression. Thus, the activation of miRNA in SCA1 brain could serve to reduce the cytotoxic effect of CAG expanded ATXN1 gene [242]. The potential use of shRNA as a therapeutic candidate in a mouse model of SCA1 was published in 2004 [243]. Intracerebellar stereotaxic injections of recombinant AAV expressing shRNA improved motor function, rescued cerebellar morphology and decreased *Atxn1* inclusions in Purkinje cells. Finally, AONs are relatively short, single-stranded DNA/RNA sequences that can bind the target mRNA through Watson–Crick base pairing. This binding arrests translation and decreases synthesis of the encoded protein. SCA2 mice treated with intrathecal AONs injections displayed a reduced cerebellar *ATXN2* expression associated with improvements in motor functions. Interestingly, Purkinje cell firing frequency returned to normal levels [244]. AONs have also been administered to SCA3 mice and were shown to sustain reduction of mutated *Atxn3* and to rescue motor impairment. A correlation was also found with a recovery of defects in Purkinje cell firing frequency [245]. In recent years, stem cell-based clinical trials have taken the emerging field offering significant potential to deliver new treatments for SCAs. The first preclinical experiment into SCA1 mouse model has been published by Kaemmerer and Low (1999). They found that transplantation of embryonic cerebellar cells led to improvement in motor skills and enhancement in Purkinje cell survival [246]. Ten years later Chintawar and his colleagues [247] transplanted neural precursor cells derived from adult mice into SCA1 mouse cerebellum. These animals functioned better on behavioral tests and displayed more surviving Purkinje cells. Another study demonstrated that cerebellar neural stem cells transplanted into cerebellum of adult SCA3 mice alleviated motor impairments [248]. In SCA2 mouse model, the intravenous injection of human mesenchymal stem cells ameliorated the survival of Purkinje cells and motor performance and delayed the onset of disease [249]. Somatic cells that can be reprogrammed into iPSCs represent the last frontier within the stem cell-based therapy [250,251]. Currently, patient-derived SCA iPSCs are tested for drug screening and disease modelling [243,252,253]. First stem cell-based clinical trials in humans with cerebellar degeneration have already tested [254,255]. The findings support cerebellar transplantation as a promising therapy but intensive preclinical research is still necessary.

## 12. Conclusions

A wide variety of mutations causing ataxia reveals the central role of cerebellar Purkinje cells in the pathogenesis of motor dysfunctions in SCAs. Almost all ionic channels or receptors expressed by Purkinje cells involved in Ca^2+^ signaling or homeostasis, when mutated, cause ataxic symptoms but, currently, how their functions are altered in disease remain to be determined. Accumulating lines of evidence suggest that multiple cellular pathways are disrupted by majority of these mutated proteins in Purkinje cells [256], but the distinctive SCA pathogenesis does not begin until Ca^2+^ signaling pathways are deranged either as a result of an toxic increase or a suppression of compensatory mechanisms likely leading to the characteristic Purkinje cells loss, cerebellar atrophy, and ataxia. Additionally, alterations of any synaptic input to the Purkinje cells also cause ataxic symptoms. In the case of the parallel fiber synapse, the reduced number of contacts between parallel fiber and Purkinje cell, the extended innervation of the climbing fiber invading the parallel fiber territory and the LTD impairment are sufficient to cause motor symptoms related to ataxia, as in GluD2 mutant mice. In contrast, even a slight maturation disturbance of the climbing fiber–Purkinje cell synapse can cause acute motor deficits suggesting that the refinement of the CF projection causes more severe ataxia than the loss of cerebellar cortex output. Notably, extracellular Ca^2+^ influx through Ca(v)2.1 is involved in development, functional differentiation and maturation of climbing fiber synapses. Studies on ataxic mutant mice have also resulted in the identification of other molecules involved in the phase of climbing fiber synapse elimination, such as PKCγ and its downstream signal transduction pathway, as well as *Atxn*1. These mechanisms mentioned above could act independently or, more likely, interact and synergize with each other, triggering the accumulation of Purkinje cell damage that eventually leads to cerebellar dysfunction. This suggests that simultaneous targeting of several pathways might be therapeutically necessary to prevent neurodegeneration and preserve neuronal function [256].

## Figures and Tables

**Figure 1 ijms-21-00216-f001:**
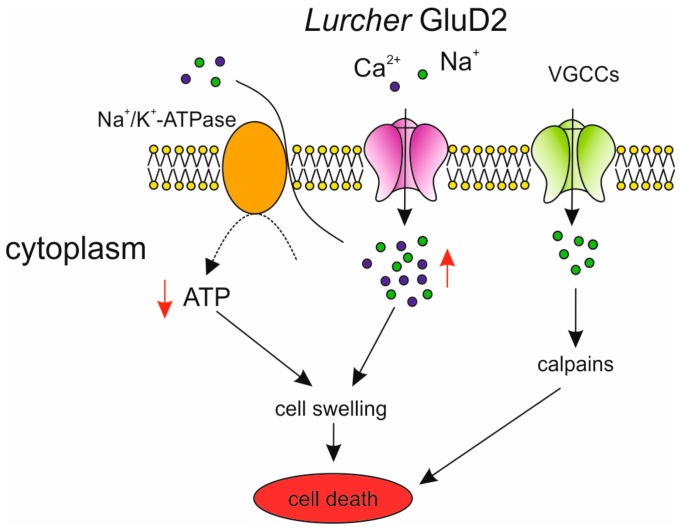
Lurcher δ2 glutamate (GluD2)-induced cell death. *Lurcher* GluD2 receptors show constitutive and continuous influx of Na^+^ and Ca^2+^. Intracellular ATP levels are decreased probably by overactivation of the Na^+^/K^+^-ATPase. Compromised ionic homeostasis together with decreased ATP levels can lead to cell swelling and subsequent cell death. Secondary Ca^2+^ influx through voltage-gated Ca^2+^ channels (VGCCs) could activate a variety of Ca^2+^-dependent enzymes, such as calpains, and potentially contribute to Purkinje cell death through different pathways. Modified from [100].

**Figure 2 ijms-21-00216-f002:**
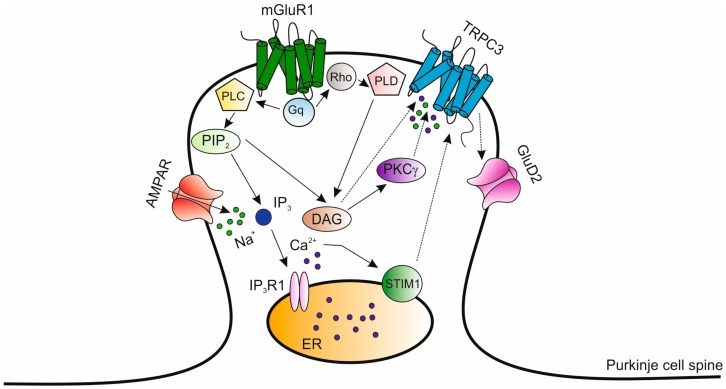
Subtype I metabotropic glutamate receptors-Transient Receptor Potential Canonical 3 (TRPC3)-PKCγ (mGluR1-TRPC3-PKCγ) signaling at Purkinje cell synapse. TRPC3 channels and PKCγ primarily respond to phospholipase C (PLC)-coupled receptors, such as mGluR1. TRPC3 channel activity is negatively regulated through phosphorylation by PKCγ. When stimulated, mGluR1 activates phospholipase C (PLC) which hydrolizes phosphatidylinositol 4,5-bisphosphate (PIP_2_) into inositol 1,4,5-trisphosphate (IP_3_) and diacylglycerol (DAG). Subsequently, the endoplasmic reticulum (ER)-Ca^2+^ store depletion by IP_3_ activates the ER Ca^2+^-sensor STIM1 which interacts with and activates TRPC3 channels. In addition, DAG has profound effects on the TRPC3 channels through PKC but it can also activate TRPC3 channels in a non-PKC-dependent manner. In turn, DAG formation is also promoted by mGluR1-dependent activation of phospholipase D (PLD) through the small GTP-binding protein Rho. In addition, mGluR1 activation triggers the opening of GluD2 receptors. Modified from [121].

**Figure 3 ijms-21-00216-f003:**
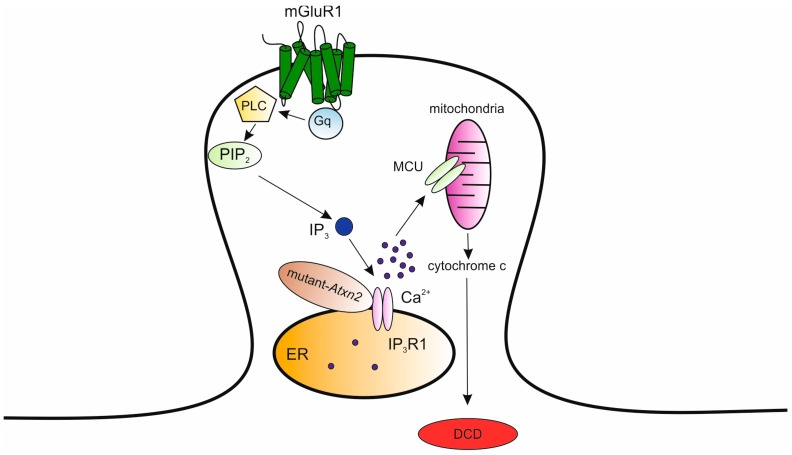
The IP_3_ receptor type 1 (IP_3_R1)-mutant *Atxn2* interaction-induced cell death. The IP_3_R1–mutant ataxin-2 (*Atxn*2) interaction results in increased IP_3_R1 activity. The abnormal Ca^2+^ signaling determines mitochondrial Ca^2+^ overload, release of cytochrome C and, consequently, induction of Purkinje cell death via dark cell degeneration (DCD). Modified from [22].

**Table 1 ijms-21-00216-t001:** SCA: Spinocerebellar ataxias.

Location	SCA	OMIM Number	Distinguishing Clinical Manifestations ^(1)^	Gene	Type of Mutation
6p22.3	SCA1	164400	Pyramidal signs, peripheral neuropathy, and ophthalmoparesis	*ATXN1*	(CAG)n
12q24.12	SCA2	183090	Hyporeflexia, tremor and slow eye movements	*ATXN2*	(CAG)n
14q32.12	SCA3	109150	Motor neuron involvement and Parkinsonian features	*ATXN3*	(CAG)n
16q22.1	SCA4	600223	Sensory peripheral neuropathy	*Unknown*	Unknown
11q13.2	SCA5	600224	Early onset and very slow disease progression.	*SPTBN2*	Point mutations
19p13.13	SCA6	183086	Late-onset, very slow disease progression. and nystagmus.	*CACNA1A*	(CAG)n
3p14.1	SCA7	164500	Visual loss	*ATXN7*	(CAG)n
13q21	SCA8	608768	Cognitive dysfunction, pyramidal and sensory signs	*ATXN8*	(CTG * CAG)n
22q13.31	SCA10	603516	Occasional epilepsy	*ATXN10*	(ATTCT)n
15q15.2	SCA11	604432	Pyramidal signs.	*TTBK2*	Point mutations
5q32	SCA12	604326	Tremor, Parkinsonian features and dementia	*PPP2R2B*	(CAG)n
19q13.33	SCA13	605259	Delayed motor and cognitive development	*KCNC3*	Point mutations
19q13.42	SCA14	605361	Dystonia and myoclonus.	*PRKCG*	Point mutations
3p26.1	SCA15/16	606658	Tremor and cognitive impairment.	*ITPR1*	Point mutations
6q27	SCA17	607136	Dementia and Parkinsonian features	*TBP*	(CAG)n
7q22–q32	SCA18	607458	Sensory and motor neuropathy	*IFRD1*	Point mutations
1p13.2	SCA19/22	607346	Cognitive impairment and myoclonus	*KCND3*	Point mutations
11q12	SCA20	608687	Cerebellar dysarthria	*Unknown*	Genomic duplication
1p36.33	SCA21	607454	Mild cognitive impairment, and Parkinsonian features	*TMEM240*	Unknown
20p13	SCA23	610245	Pyramidal signs	*PDYN*	Point mutations
2p21–p13	SCA25	608703	Peripheral neuropathy,	*Unknown*	Unknown
19p13.3	SCA26	609306	Eye movement abnormalities.	*EEF2*	Point mutations
13q33.1	SCA27	609307	Tremor and dystonia	*FGF14*	Point mutations
18p11.21	SCA28	610246	Spastic ataxia	*AFG3L2*	Point mutations
3p26.1	SCA29	117360	Intellectual disability.	*ITPR1*	Point mutations
4q34.3–q35.1	SCA30	613371	Pure ataxia.	*ODZ3*	Unknown
16q21	SCA31	117210	Abnormal sensation	*BEAN1*	(TGGAA)n
6q14.1	SCA34	133190	Hyperkeratosis	*ELOVL4*	Unknown
20p13	SCA35	613908	Ocular dysmetria, tremor and hyperreflexia	*TGM6*	Point mutations
20p13	SCA36	614153	Motor neuron involvement	*NOP56*	(GGCCTG)n
1p32.2	SCA37	615945	Altered vertical eye movements.	*DAB1*	(GGCCTG)n
6p12.1	SCA38	615957	Nystagmus and dysarthria	*ELOVL5*	Point mutations
14q32.11–q32.12	SCA40	616053	Ocular dysmetria and tremor	*CCDC88C*	Point mutations
4q27	SCA41	616410	Imbalance and loss of coordination	*TRPC3*	Point mutations
17q21.33	SCA42	618087	Gait instability, dysarthria and nystagmus	*CACNA1G*	Point mutations
3q25.2	SCA43	617018	Peripheral neuropathy	*MME*	Point mutations
6q24.3	SCA44	617691	Dysarthria, dysphagia and dysmetria	*GRM1*	Point mutations
5q33.1	SCA45	617769	Nystagmus, and dysarthria.	*FAT2*	Point mutations
19q13.2	SCA46	617770	Sensory ataxic neuropathy	*PLD3*	Point mutations
12p13.31	DRLPA	125370	Involuntary movements, mental and emotional problems	*ATN1*	(CAG)n
4q22.1–q22.2	GRID2-related spinocerebellar ataxia	616204	Motor, speech and cognitive delay and eye movement abnormalities	*GRID2* *Rarely AD inheritance*	Point mutations

^(^^1)^ from https://www.orpha.net/consor/cgi-bin/index.php?lng=EN or https://www.omim.org/.

**Table 2 ijms-21-00216-t002:** Features of SCAs linked with abnormal Ca^2+^ signaling.

SCA	Gene	Protein	Effect on Ca^2+^ Signaling
SCA1	*ATXN1*	Ataxin-1	Decrease
SCA2	*ATXN2*	Ataxin-2	Increase
SCA3	*ATXN3*	Ataxin-3	Increase
SCA6	*CACNA1A*	Ca^2+^voltage-gated channel subunit α1A	Decrease
SCA14	*PRKCG*	PKCγ	Increase/Decrease
SCA15/16	*ITPR1*	IP_3_ receptor	Increase/Decrease
SCA29	*ITPR1*	IP_3_ receptor	Decrease
SCA41	*TRPC3*	TRPC3 channel	Increase
SCA42	*CACNA1G*	Ca^2+^ voltage-gated channel subunit α1G	Decrease
SCA44	*GMR1*	mGlu receptor 1	Increase

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
