# Peer review of "Disrupted Calcium Signaling in Animal Models of Human Spinocerebellar Ataxia (SCA)"

_ijms, 2019, doi:10.3390/ijms21010216_

Round 1
Reviewer 1 Report
The Review is interesting and extensive in evaluating the different causes of SCA, that share abnormality in Ca2+ flux in the pathogenesis. A few issues should be addressed:
Comments
The Authors should give the prevalence of the different forms of SCA. The Table with the etiologies of SCA seems to the Reviewer not strictly necessary. It could be added as supplemental material or simplified. An information that could be useful is to focus only on the distinctive features (functional or clinical) of each genetic alteration (e.g. SCA4 sensory neuropathy). Another way to simplify could be to group the distinct disorders for selected clinical or functional parameters (e.g. cognitive dysfunction); The Ca2+ flux is a common biochemical event shared by several signaling pathways, as correctly underlined by the Authors; however, an attempt to tie together the individual gene alteration with a specific functional alteration or biologic process should be made in an easier and more accessible way for the reader. As examples: mutation in ataxin-2 leads to abnormal autophagy or mutation in Itpr1 leads to abnormal in cell signaling, etc. A table easy to read could help; The Authors could include a short chapter on therapeutic approaches.Author Response
We thank the reviewer for comments that helped focusing better on important aspects of the review.
1.The Table with the etiologies of SCA (Table 1) has been simplified focusing on the distinctive clinical manifestations (please see attachment).
2. The link between features of SCAs and effects on calcium signaling were summarized in a new Table (Table.2)
3. A short chapter on therapeutic approaches has been added.
Table.1 Autosomal Dominant Spinocerebellar ataxias: mutations and clinical manifestations
from https://www.orpha.net/consor/cgi-bin/index.php?lng=EN or https://www.omim.org/
Reviewer 2 Report
This is an excellent review of SCAs, with an emphasis on genes involving calcium metabolism which is proved to be vital to the pathogenesis of SCAs. The summary of the genes and the pathways, especially the graphical abstract, is excellent. This reviewer suggest direct acceptance for publication, however, if we need to pick up some weak points, which does not reduce the merit of this manuscript, will be the authors are advised, but not mandatory, to also summarize from the literature or from their own group, that the reported animal models of SCAs, since the authors had stated that in the abstract, since it is an excellent review, some space to cover this aspect will be welcome.Author Response
We thank the Reviewer for the extremely positive comments on this review.
We have added a new Table linking features of SCAs with effects on calcium signaling and short chapter on therapeutic approches. Moreover, Table 1 has been slightly modified to make it more accessible to the reader.
We believe that these changes can make the text more in line with the abstract.